# The Effects of a Tailored Dietary Education Program for Older Adult Patients on Hemodialysis: A Preliminary Study

**DOI:** 10.3390/healthcare11172371

**Published:** 2023-08-22

**Authors:** Gayoung Park, Seunghye Choi

**Affiliations:** 1Yeokgok Medihols Medical Center, 567, Gyeongin-ro, Bucheon-si 14727, Republic of Korea; gy4573@gachon.ac.kr; 2College of Nursing, Gachon University, 191, Hambangmoe-ro, Incheon 21936, Republic of Korea

**Keywords:** self-efficacy, nutrition, hemodialysis, older adults

## Abstract

This study aimed to investigate the overall effects of a tailored dietary education program for older adult patients on hemodialysis (HD) based on self-efficacy theory, dietary knowledge and habits, nutritional intake, and biochemical parameters. A nonequivalent control group pre-test–post-test design was conducted for 8 weeks. The experimental and control groups received a weekly nutritional program and standard nursing care with an additional educational session, respectively. A clinical survey was conducted before and after the intervention. After the intervention, self-efficacy, dietary knowledge, and dietary habits were higher in the experimental group than in the control group. Moreover, carbohydrate, phosphorus, and sodium intake significantly decreased post-intervention in the experimental group but not in the control group. The dietary education program for older HD patients showed positive effects on boosting their self-efficacy, increasing dietary knowledge, improving dietary habits, and decreasing carbohydrate, calcium, phosphorus, and sodium intake.

## 1. Introduction

Chronic kidney disease is a progressive condition characterized by structural and functional changes in the kidney that may occur due to various underlying conditions [1]. Renal replacement therapies such as hemodialysis (HD), peritoneal dialysis (PD), or kidney transplantations are life-sustaining treatments for people with kidney failure [1]. According to 2020 statistics, 81%, 3.9%, and 15.1% of patients with chronic renal failure in Korea received HD, PD, and kidney transplantations, respectively [2]. The proportion of older adult patients on HD aged > 65 years has increased every year, rising rapidly from 54.6% in 2019 to 63.7% in 2020 [2].

Individuals with chronic kidney disease, particularly those undergoing maintenance dialysis, are prone to protein-energy wasting. Additionally, dietary education is also important [3]. Moreover, helping patients manage their diet not only positively affects their nutritional status, but also reduces the morbidity and mortality associated with complications such as heart disease [4]. Patients undergoing HD need to balance the etiology that leads to kidney disease (usually hypertension or diabetes) with the high protein and low fluid, sodium, phosphorus, and potassium dietary requirements of renal replacement therapy [5].

Multiple dietary restrictions recommended for older adult patients on HD may lead to a poor dietary status [6]. For the early identification and treatment of patients with a poor dietary status, multiple evaluations of their nutritional condition using various nutritional assessments, including anthropometric, biochemical, and dietary methods, are needed [7].

Adherence to a dietary program is crucial; however, the diet prescribed during HD is difficult to follow, especially for older adults [8]. According to a previous study, after implementing a tailored dietary education program consisting of 30 min for each session for 4 weeks, there was a significant increase in HD patients’ diet knowledge, diet self-care compliance, and self-efficacy [9]. Moreover, self-efficacy was associated with better adherence to the dietary program [10]. Despite the increase in the number of older adult patients receiving HD, dietary interventions based on self-efficacy theory are rare. Therefore, this study aimed to develop a dietary education program based on self-efficacy theory for older adult patients undergoing HD and to evaluate its effects on patients’ self-efficacy, dietary knowledge, dietary habits, nutrient intake, biochemical markers, and anthropometric changes.

## 2. Materials and Methods

### 2.1. Study Design

This study was designed with a nonequivalent control group pre-test–post-test design.

### 2.2. Setting

This study was conducted at a clinic center in Gyeonggi-do, Korea, from January 2022 to March 2022. A recruitment bulletin was posted for older adult patients registered at the clinic who were receiving HD. 

### 2.3. Participants

The inclusion criteria were older adult patients aged ≥ 65 years who had been on HD for 3 months or more, had an arteriovenous fistula, understood the purpose and method of the study and the contents of education, provided written consent to participate, and communicated verbally. The sample size was calculated using the G*power 3.1.9.2 program [11]. The total sample size required was 28 in each group for the independent *t*-test, α = 0.05, power (1 − β) = 0.90, and effect size of 0.8 (large effect size). Considering the dropout rate was 5–10%, a total of 60 subjects were recruited, including 30 subjects and 30 control subjects. Patients with an instability due to serious heart disease, liver disease, cancer, or a restrictive diet were excluded. Sixty patients were screened and allocated to the experimental and control (n = 30 each) groups according to the dialysis schedule (morning time vs. afternoon time). In the experimental and control groups, dropouts owing to hospitalization (n = 2 and n = 1, respectively) and temporary quarantine for coronavirus infection (n = 3 and n = 4, respectively) occurred. Hence, 25 participants each in the experimental and control groups were included in the final analysis. 

### 2.4. Ethical Considerations

This study was approved by the Institutional Review Board of G University (1044396-202112-HR-244-01) and conducted in accordance with the Declaration of Helsinki. In addition, before starting the study, we obtained permission from the Head of the institution. All participants were provided with a form that explained the background and purpose of the study, survey content, benefits of participation, confidentiality, and right to withdraw from the study. Routine nursing care was provided to the control group during the study period, and a dietary education leaflet and nutrition counseling were provided post-study. All participants were given a small gift before and after the intervention.

### 2.5. Study Procedure

This study used an 8-week quasi-experimental design. As this study was conducted at one center, there was a risk of diffusion or imitation of treatments in the educational content between the experimental group and control group. Therefore, instead of a random assignment, elderly HD patients receiving dialysis in the afternoon were assigned to the experimental group and patients receiving dialysis in the morning were assigned to the control group. We used a dietary education program based on self-efficacy for older adult patients on HD, which consisted of verbal persuasion, improvements in physical and emotional states, mastery experience, and social modeling (Table 1) [12]. For educational material, we used Microsoft PowerPoint to create interactive presentations to increase the understanding of older adult patients. A large font size and clear photos and pictures were used. Difficult words or medical terms were replaced with commonly used words for easy comprehension by patients. Two head nurses, a chief nurse, a nephrologist, and a nursing professor were consulted on the validity of the program contents, and the CVI (content validity index) was found to be 0.8–1.0 for all items. 

The experimental group received individual dietary education for a total of eight sessions, 20–30 min per week. Education was provided during or before and after dialysis; an artificial kidney was used as a teaching aid. Coronavirus disease 2019 (COVID-19) guidelines were followed by the researchers during all the sessions. 

### 2.6. Instruments

#### 2.6.1. General Characteristics of the Subjects

The participants were interviewed using a structured questionnaire that included sex, age, spouse (yes/no), education level, economic status (high, > 250 million; middle, 1–250 million; low, < 1 million), religion (yes/no), duration of HD, route of HD (arteriovenous fistula/graft fistula), comorbidity (yes/no), medication (yes/no), and experience of diet education (yes/no).

#### 2.6.2. Self-Efficacy

Self-efficacy is defined as an individual’s belief in their ability to exhibit a particular behavior, even in a challenging situation [15]. We used a specific self-efficacy scale for patients on HD [16]. It consisted of a 4-point Likert scale, with higher scores indicating greater self-efficacy (9–36 points; Cronbach’s α = 0.77). In this study, Cronbach’s ⍺ was 0.694. 

#### 2.6.3. Hemodialysis Dietary Knowledge

Hemodialysis dietary knowledge of patients on HD was measured with a nutritional knowledge tool, which was developed by Seo [17] and modified by Na [18]. Hemodialysis dietary knowledge comprised 12 questions regarding the rationales and principles of the hemodialysis diet. The total score ranged from 0 to 12, with higher scores indicating greater knowledge of the HD diet. In a previous study that surveyed older adult patients on HD, Cronbach’s ⍺ was 0.89 [18]. In this study, Cronbach’s α was 0.73. 

#### 2.6.4. Dietary Habits

In this study, dietary habits of the enrolled patients were assessed based on the factors of dietary habits specifically appropriated for patients on HD [17]. It consisted of questions regarding the desired food intake (five items) and general dietary intake (two items). Subsequently, 0 (0–2 days), 1 (3–5 days), and 2 points (6–7 days) were given according to the number of practices for one week. The total score was 14 points; higher scores indicated better dietary habits. 

#### 2.6.5. Nutritional Intake

Twenty-four-hour food recall methods were used to obtain data on nutrient intake and food consumption, before and after interventions. The participants were asked to record all food and drink consumed over 1–2 days, and the researchers confirmed all food records. Moreover, the average daily nutrient intake (total calories, carbohydrate, lipid, protein, fiber, calcium, phosphorus, sodium, potassium, and magnesium) was calculated using the computer-aided nutrition analysis program (CAN Pro) 5.0 (The Korean Nutrition Society, Seoul, Korea) [19]. According to the dietary guidelines for hemodialysis patients, a daily intake of phosphorus less than 800 mg was defined as appropriate, less than 800–1000 mg as borderline, and more than 1000 mg as excessive intake, while a daily intake of sodium and potassium less than 2000 mg was appropriate, and more than this was excessive intake [20]. 

#### 2.6.6. Biochemical and Anthropometric Parameters

We reviewed patients’ medical records such as serum potassium (mEq/L) and phosphorus (mg/mL) levels and weight gain between HD sessions. Serum potassium and phosphorus levels were examined as a result of routine tests performed at the beginning of each month immediately before initiating HD. These values were measured by the Department of Laboratory Medicine at the hospital by collecting 6 cc of blood. The weight gain between HD sessions was calculated using a scale by measuring the difference between the weight before and after HD (DW-150, CAS, KOREA).

### 2.7. Data Analysis

Statistical analyses were performed using SPSS 24.0 (IBM Corp., Armonk, NY, USA). Cronbach’s α was used to assess the reliability of the scale. The general characteristics of the experimental and control groups were investigated using a *t*-test or a χ^2^-test. Self-efficacy, hemodialysis dietary knowledge, dietary habits, nutritional intake, and biochemical and anthropometric parameters were evaluated using the Kolmogorov–Smirnov normality test followed by the *t*-test or Mann–Whitney U test. Within- or between-group effects of the dietary education program for older adult patients on HD based on self-efficacy theory were evaluated using the difference between pre- and post-intervention values. After the Kolmogorov–Smirnov normality test, pre- and post-intervention data were compared using a *t*-test or Mann–Whitney U test to assess between-group differences and a paired *t*-test or Wilcoxon test for within-group changes. The total calorie and energy intake fractions by source ratio were analyzed as means, standard deviations, and percentages. The level of statistical significance was set at *p* < 0.05.

## 3. Results

### 3.1. Homogeneity Test of Participants’ General Characteristics

In a homogeneity test that was conducted, the two groups did not differ in terms of their general characteristics (Table 2). We also conducted a homogeneity test of pre-intervention variables, including self-efficacy, hemodialysis dietary knowledge, dietary habits, biochemical parameters, and nutritional intake; the two groups did not differ in these variables, except dietary habits (*p* = 0.018) (Table 3). The pre-intervention dietary habit score was higher in the control group than in the experimental group (Table 3).

### 3.2. Differences in Self-Efficacy, Hemodialysis Dietary Knowledge, Dietary Habits, Biochemical Parameters, and Nutritional Intake

#### 3.2.1. Group Differences in Self-Efficacy, Hemodialysis Dietary Knowledge, Dietary Habits, Biochemical Parameters, and Nutritional Intake

In the homogeneity test before the intervention, the control group had a higher dietary habit score than that of the experimental group; therefore, we compared the values obtained by subtracting the post- and pre-intervention scores to confirm the effectiveness of the program. Self-efficacy, dietary knowledge, and dietary habits were higher in the experimental group than in the control group (*p* = 0.001, *p* = 0.010, and *p* = 0.003, respectively) (Table 4). In contrast, the intake of carbohydrate, calcium, phosphorus, sodium, and magnesium was lower in the experimental group than in the control group (*p* = 0.020, *p* = 0.009, *p* = 0.014, *p* = 0.020, and *p* = 0.015, respectively) (Table 4). 

#### 3.2.2. Within-Group Changes in Self-Efficacy, Hemodialysis Dietary Knowledge, Dietary Habits, Biochemical Parameters, and Nutritional Intake before and after Intervention

In the experimental group, self-efficacy, hemodialysis dietary knowledge, and dietary habits increased and serum potassium levels decreased post-intervention (*p* < 0.001, *p* < 0.001, *p* < 0.001, and *p* = 0.015, respectively) (Table 4). In contrast, in the control group, dietary knowledge (*p* = 0.003) and calcium intake increased post-intervention (*p* = 0.038) (Table 4). 

### 3.3. Total Calorie Intake and Fraction of Energy Intake by Source

As the estimated energy requirement (EER) differs according to age and sex, the total calorie intake and fraction of energy intake by source according to the age and sex of the subjects are presented in Table 4.

### 3.4. Differences in Excessive Intake of Phosphorus, Sodium, and Potassium

After the intervention, the rates of excessive intake of phosphorus and sodium were 8.0% and 72% in the experimental group and 32% and 88% in the control group, respectively. Compared with before the experiment, in the experimental group, the excessive intake of phosphorus decreased from 24% to 8% and that of sodium decreased from 84.0% to 72%. However, in the control group, the excessive intake of phosphorus slightly decreased from 36% to 32%, but the rate of sodium intake increased from 72% to 88%. The rate of excessive intake of potassium decreased from 48% to 28% in the experimental group. There was no significant difference between the groups in the rate of excessive intake of potassium before the intervention, but there was a significant difference after the intervention (*p* = 0.042) (Table 5).

## 4. Discussion

This study aimed to identify the effects of a dietary education program based on self-efficacy theory for older adult patients on HD that included self-efficacy, hemodialysis dietary knowledge, dietary habits, biochemical parameters, and nutritional intake in Korean older adult patients on HD. The overall energy intake of the participants tended to be insufficient compared with the dietary reference intake for the Korean population [21]. The EER is 2000 kcal for men over 65 years, 1600 kcal for women 65–74 years, and 1500 kcal for those older than 75 years [21]. Among the study population, only women aged > 75 years in the experimental group had adequate energy intake; the rest of the experimental and control groups had lesser energy intake than the EER [21]. Maintenance HD not only induces a catabolic status in patients [22], but also causes loss of appetite, loss of serum albumin-induced HD, comorbid conditions, and enhanced protein degradation that is observed with aging [23]; a poor nutritional status and protein-energy wasting are common among patients on maintenance HD [24]. Older adult patients on HD commonly have comorbidities. Additionally, the HD diet is difficult to follow, especially for older adults [8]. Therefore, it is important to develop individualized nutritional intervention programs.

The recommended energy ratios for HD patients are 50%, 15%, and 35% for carbohydrate, protein, and fat, respectively [25]. However, the energy ratios of the experimental and control groups in the present study were greater than 60%, 12–15%, and 10–20%, respectively; for instance, the carbohydrate intake was higher and the fat intake was lower than the recommended ratio. Fat intake is important for the maintenance of total energy intake and according to a previous study, the serum cholesterol level and mortality risk among patients on HD showed a U-shaped curve [7]. In this dietary education program, customized counseling was provided to the participants based on pre-test data so that they could meet the appropriate energy intake fraction. 

In the present study, the experimental group had significantly higher self-efficacy, dietary knowledge, and dietary habits and significantly lower carbohydrate, calcium, phosphorus, and sodium intake post-intervention compared with the control group. There was no significant difference in the blood chemistry tests between the groups; however, the serum potassium level in the experimental group was significantly lower post-intervention than pre-intervention. The program showed positive effects, probably because it was designed based on self-efficacy, relying on behavioral changes induced by building mastery experiences, emphasizing past successes, social modeling, improving emotional states, and verbal persuasion. These results were consistent with a previous study, which applied sodium reduction to the nutritional intervention program for patients on HD [8]. Dietary interventions designed to improve self-efficacy also positively affected the dietary sodium density reduction, especially in groups with low baseline self-efficacy [8]. Self-efficacy is a crucial mediator between knowledge and self-care [26]; therefore, strategies to increase self-efficacy are essential for the promotion of adherence to dietary guidelines among vulnerable older adult populations. 

We found that sodium and phosphorus intake significantly decreased in the experimental group compared with those in the control group after the program. According to a previous study, a high phosphorus intake is known to increase the mortality rate among patients on HD [27]. Moreover, an excessive sodium intake may affect the elevation of blood pressure in these patients [28]. Therefore, the present program effectively reduced sodium and phosphorus intake in patients on HD. The positive changes in nutrient intake despite the relatively short intervention period (8 weeks) may have been owing to the application of personalized interventions. 

A high potassium intake is known to increase the mortality rate of patients on HD [27]; therefore, a nutritional intervention program that could reduce the potassium intake in HD patients is very crucial. In the present study, no significant difference was noted between the serum potassium levels in the experimental and control groups. However, the dietary potassium intake and serum potassium level decreased in the experimental group post-intervention. Koreans tend to prefer food containing large amounts of potassium and it is difficult to reduce potassium intake in a short period. Therefore, it is necessary to educate patients on changing cooking methods for food containing high levels of potassium [29]. As potassium is a water-soluble substance, it can be dissolved in water by peeling and boiling vegetables rather than baking or frying them [30]. A long-term intervention is needed to modify the choice of low-potassium-containing food and cooking methods. 

Serum phosphorus levels and weight gains of the participants did not significantly change in either the experimental or control groups. However, according to a previous study, a high phosphorus intake increased the mortality rate of patients on HD, despite adjusting for serum phosphorus levels and potassium and protein intake [27]. In our study, there was no significant change in the weight of the participants before and after HD initiation. This could be attributed to the fact that the study was conducted in winter; hence, the amount of exercise may have decreased among older adults, particularly in those who were chronically ill who reduced outdoor activities due to the COVID-19 pandemic. 

Our study had several limitations. First, this study was conducted during the COVID-19 pandemic, owing to which the dropout rate was higher than expected. Second, this study was conducted at only one clinic, so our results cannot be generalized to all older adult patients on HD in Korea.

## 5. Conclusions

In conclusion, a dietary education program for older adult patients on HD based on self-efficacy theory positively affected self-efficacy and dietary knowledge, improved dietary habits, and decreased carbohydrate, calcium, phosphorus, and sodium intake among these patients. Future research should focus on the long-term effects of dietary education programs based on self-efficacy in older adult patients undergoing HD.

## Figures and Tables

**Table 1 healthcare-11-02371-t001:** Dietary education program based on self-efficacy theory for older adult patients undergoing hemodialysis.

Session	Source of Self-Efficacy	Contents of the Intervention
0		Participant screening and pre-test
1	Verbal persuasion	Introduced the program and planned with the participants
	Improving emotional states	Provided face-to-face counseling to participants for pre-test results Interviewed regarding the 24 h food recall diary Provided a small gift
2	Verbal persuasion	Educated participants regarding diet prescribed for patients on hemodialysis using computer-aided visual data: Section 1. Restriction of Sodium and Water [13,14] Provided a leaflet and small stickers that could be attached to a refrigerator and encouraged the use of stickers when following dietary guidelines
3	Verbal persuasion	Provided information to participants about their current nutrient intake and status Provided exercise routines based on their weight
	Improving emotional states	Verified current dietary habits and barriers to their diet Shared the difficulty among other participants and provided emotional support
	Mastery experience	Reviewed previous educational content and checked their compliance with the leaflet
4	Verbal persuasion	Educated participants regarding diet prescribed for patients undergoing hemodialysis using computer-aided visual data: Section 2. Desirable Hemodialysis Life [13,14]
	Mastery experience	Reviewed the previous educational content and checked their compliance with the leaflet Encouraged participants to express their achievement and provided a small gift
5	Verbal persuasion	Educated participants regarding the diet prescribed for patients on hemodialysis using computer-aided visual data: Section 3. Restriction of Potassium and Phosphorus [13,14] Provided small stickers and encouraged participants to set goals according to the diet guidelines specified for patients on hemodialysis
	Mastery experience	Reviewed previous educational content and checked their compliance with the leaflet Encouraged participants to express their achievements
6	Verbal persuasion	Educated participants on diet specified for patients on hemodialysis using computer-aided visual data: Section 4. Healthy Diet [13,14]
	Mastery experience	Reviewed previous educational content and checked their compliance with the leaflet Encouraged participants to express their achievements
7	Verbal persuasion	Inspected participants’ diet by checking their shopping list and reinforced the importance of the diet prescribed for patients with chronic kidney disease
	Mastery experience	Reviewed the previous educational content and checked their compliance with the leaflet
8	Improving physical and emotional states	Performed post-test and provided a small gift
	Mastery experience	Compared pre- and post-test results Encouraged participants to express their achievements
	Social modeling	Introduced successful cases among participants

**Table 2 healthcare-11-02371-t002:** Homogeneity test of participants’ general characteristics.

Variables	Category	Exp. (n = 25) N (%)	Con. (n = 25) N (%)	χ^2^	*p*-Value
Sex	Male	12 (44.4)	15 (55.6)	0.725	0.395
Age (years)	65–74	14 (48.3)	15 (51.7)	0.082	0.774
	≥75	11 (52.4)	10 (47.6)		
Spouse	Yes	19 (48.7)	20 (51.3)	0.117	0.733
	No	6 (54.5)	5 (45.5)		
Education level (years)	Less than middle school	15 (55.6)	12 (44.4)	0.725	0.395
	High school graduate or higher	10 (43.5)	13 (56.5)		
Economic status	High	6 (60.0)	4 (40.0)	2.133	0.344
	Middle	16 (53.3)	14 (46.7)		
	Low	3 (30.0)	7 (70.0)		
Religion	Yes	13 (37.1)	22 (62.9)	7.714	0.005
	No	12 (80.0)	3 (20.0)		
Duration of HD (years)	>5	7 (53.8)	6 (46.2)	0.331	0.848
	5–10	5 (55.6)	4 (44.4)		
	<10	13 (46.4)	15 (53.6)		
Route of HD	AVF	20 (50.0)	20 (50.0)	0.000	1.000
	Graft fistula	5 (50.0)	5 (50.0)		
Comorbidity	Yes	24 (50.0)	24 (50.0)	0.000	1.000
	No	1 (50.0)	1 (50.0)		
Medication	Yes	25 (51.0)	24 (49.0)	1.02	0.312
	No	0 (0.0)	1 (100.0)		
Smoking	Yes	2 (100.0)	0 (0.0)	2.083	0.245
	No	23 (47.9)	25 (52.1)		
Drinking	Yes	4 (50.0)	4 (50.0)	0.000	0.649
	No	21 (50.0)	21 (50.0)		
Experience of diet education	Yes	3 (27.3)	8 (72.7)	2.914	0.088
	No	22 (56.4)	17 (43.6)		

Exp: experimental group; Con: control group; HD: hemodialysis; AVF: arteriovenous fistula.

**Table 3 healthcare-11-02371-t003:** Homogeneity test of self-efficacy, hemodialysis dietary knowledge, dietary habits, biochemical parameters, and nutritional intake.

Variable	Unit	Exp. (n = 25) Mean (SD)	Con. (n = 25) Mean (SD)	*t*	*p*-Value
Self-efficacy		25.48 (4.43)	25.48 (5.95)	288.00 *	0.633
Dietary knowledge		5.84 (2.84)	7.00 (2.84)	−1.444	0.155
Dietary habits		9.60 (1.68)	10.80 (1.78)	−2.449	0.018
*Biochemical parameters (serum level)*					
Potassium	mEq/L	5.11 (0.78)	5.46 (0.70)	−1.681	0.099
Phosphorus	mg/ml	4.14 (1.18)	4.47 (1.20)	−0.998	0.323
Weight gain	kg	2.56 (0.96)	2.74 (2.22)	300.00 *	0.808
*Nutrient intake*					
Calories	kcal	1510.51 (337.21)	1559.07 (475.87)	−0.416	0.679
Carbohydrate	g	249.54 (56.95)	241.88 (76.75)	0.401	0.690
Lipid	g	30.46 (16.28)	37.70 (17.72)	230.00 *	0.109
Protein	g	51.98 (13.15)	54.75 (20.61)	−0.567	0.573
Fiber	g	17.43 (6.14)	18.78 (7.44)	−0.701	0.487
Calcium	mg	408.97 (209.69)	378.57 (201.53)	0.523	0.604
Phosphorus	mg	827.85 (224.09)	840.84 (322.46)	−0.165	0.869
Sodium	mg	3091.39 (1186.96)	3133.25 (1451.17)	−0.112	0.912
Potassium	mg	1973.89 (648.10)	2158.75 (868.82)	−0.853	0.398
Magnesium	mg	76.24 (42.88)	96.51 (66.05)	259.00 *	0.299

Exp: experimental group; Con: control group; SD: standard deviation; * Mann–Whitney U test.

**Table 4 healthcare-11-02371-t004:** Differences in self-efficacy, hemodialysis dietary knowledge, dietary habits, biochemical parameters, and nutritional intake.

Variable	Group	Pre-Test	Post-Test	Paired *t* or z	*p*-Value	Differences (Pre–Post)	*t* or U	*p*-Value
		Mean (SD)	Mean (SD)			Mean (SD)		
Self-efficacy	Exp.	25.48 (4.43)	29.20 (4.05)	−3.647 ^†^	< 0.001	−3.72 (3.82)	−3.376	0.001
	Con.	25.48 (5.95)	24.84 (6.61)	−0.686 ^†^	0.493	0.64 (5.20)		
Dietary knowledge	Exp.	5.84 (2.84)	8.96 (1.88)	−3.835 ^†^	< 0.001	−3.12 (2.64)	180.50 *	0.010
	Con.	7.00 (2.84)	8.32 (2.15)	−2.942 ^†^	0.003	−1.32 (1.75)		
Dietary habits	Exp.	9.60 (1.68)	11.24 (1.23)	−4.389	< 0.001	−1.64 (1.87)	164.00 *	0.003
	Con.	10.80 (1.78)	10.76 (1.23)	0.102	0.920	0.04 (1.97)		
*Serum level*								
Potassium	Exp.	5.11 (0.78)	4.77 (0.66)	2.634	0.015	0.34 (0.65)	0.782	0.439
	Con.	5.46 (0.70)	5.30 (0.85)	0.889	0.383	0.16 (0.92)		
Phosphorus	Exp.	4.14 (1.18)	4.58 (0.96)	−1.782	0.087	0.44 (1.23)	0.099	0.921
	Con.	4.47 (1.20)	4.88 (.088)	−1.517	0.142	0.40 (1.33)		
Weight gain	Exp.	2.56 (0.96)	2.47 (1.08)	−0.592 ^†^	0.554	0.08 (0.52)	−1.015	0.315
	Con.	2.74 (2.22)	2.26 (0.76)	−0.943 ^†^	0.346	0.49 (1.91)		
*Nutrient intake*								
Calories	Exp.	1510.51 (337.21)	1402.08 (297.53)	−1.117 ^†^	0.264	108.43 (441.96)	247.00 *	0.204
	Con.	1559.07 (475.87)	1669.15 (606.08)	−0.767 ^†^	0.443	−110.08 (602.36)		
Carbohydrate	Exp.	249.54 (56.95)	222.69 (48.46)	1.910	0.068	26.85 (70.27)	193.00 *	0.020
	Con.	241.88 (76.75)	254.70 (75.29)	−0.724	0.476	−12.82 (88.57)		
Lipid	Exp.	30.46 (16.28)	32.11 (14.71)	−1.440 ^†^	0.150	−1.64 (20.75)	263.00^*^	0.337
	Con.	37.70 (17.72)	40.63 (22.66)	−0.202 ^†^	0.840	−2.93 (20.82)		
Protein	Exp.	51.98 (13.15)	48.13 (13.33)	1.040	0.309	3.85 (18.50)	1.942	0.058
	Con.	54.75 (20.61)	63.37 (28.55)	−1.643	0.114	−8.62 (26.23)		
Fiber	Exp.	17.43 (6.14)	16.00 (4.18)	0.977	0.338	1.43 (7.32)	1.382	0.173
	Con.	18.78 (7.44)	20.46 (9.91)	−0.983	0.335	−1.68 (8.57)		
Calcium	Exp.	408.97 (209.69)	317.53 (151.32)	1.612	0.120	91.45 (283.67)	2.702	0.009
	Con.	378.57 (201.53)	509.45 (295.84)	−2.196	0.038	−130.88 (297.94)		
Phosphorus	Exp.	827.85 (224.09)	710.00 (208.75)	1.786	0.087	117.86 (330.03)	2.542	0.014
	Con.	840.84 (322.46)	976.57 (440.11)	−1.815	0.082	−135.73 (373.88)		
Sodium	Exp.	3091.39 (1186.96)	2604.12 (804.60)	−1.359 ^†^	0.174	487.26 (1522.85)	193.00 *	0.020
	Con.	3133.25 (1451.17)	3811.89 (1852.60)	−1.951 ^†^	0.051	−678.64 (1520.42)		
Potassium	Exp.	1973.89 (648.10)	1688.44 (577.00)	1.522	0.141	1935.29 (647.72)	−0.676	0.503
	Con.	2158.75 (868.82)	2301.12 (1027.20)	−0.898	0.378	2081.90 (870.42)		
Magnesium	Exp.	76.24 (42.88)	76.84 (41.56)	−0.040 ^†^	0.968	−1612.20 (589.67)	2.546	0.015
	Con.	96.51 (66.05)	113.01 (55.30)	−1.224 ^†^	0.221	−2204.60 (1002.99)		

Exp: experimental group; Con: control group; SD: standard deviation; * Mann–Whitney U test; ^†^ Wilcoxon test.

**Table 5 healthcare-11-02371-t005:** Differences in excessive intake of phosphorus, sodium, and potassium.

Variable	Category	Pre-Test				Post-Test			
		Exp. (n = 25) N (%)	Con. (n = 25) N (%)	χ^2^	*p*-Value	Exp. (n = 25) N (%)	Con. (n = 25) N (%)	χ^2^	*p*-Value
Phosphorus	Appropriate	13 (52.0)	13 (52.0)	1.600	0.449	15 (60.0)	11 (44.0)	4.501	0.105
	Borderline	6 (24.0)	3 (12.0)			8 (32.0)	6 (24.0)		
	Excess	6 (24.0)	9 (36.0)			2 (8.0)	8 (32.0)		
Sodium	Appropriate	4 (16.0)	7 (28.0)	1.049	0.248	7 (28.0)	3 (12.0)	2.000	0.145
	Excess	21 (84.0)	18 (72.0)			18 (72.0)	22 (88.0)		
Potassium	Appropriate	13 (52.0)	10 (40.0)	0.725	0.285	18 (72.0)	11 (44.0)	4.023	0.042
	Excess	12 (48.0)	15 (60.0)			7 (28.0)	14 (56.0)		

Exp: experimental group; Con: control group.

## Data Availability

The datasets generated and/or analyzed during the current study are not publicly available to protect the participants, but are available from the corresponding author upon reasonable request.

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
