# Peer review of "The Effects of a Tailored Dietary Education Program for Older Adult Patients on Hemodialysis: A Preliminary Study"

_healthcare, 2023, doi:10.3390/healthcare11172371_

Round 1

Reviewer 1 Report

Authors have developed an intervention strategy that improves HD patients’ knowledge and behaviour based on self-efficacy theory. There are several issues should be addressed before considering publication.

First, authors should provide a more thorough literature review on the intervention studies for HD patients rather than simply saying there are rare attempts based on self-efficacy theory. What other strategies have been used? Have they used other behavioral theories? What is the study duration and sample size of previous studies? That would be important for readers to understand the current development in this research area.

Second, the use of quasi-experimental design should be justified. Since individual dietary education was administered, a RCT appeared to be more reasonable. The problem of quasi-experimental design has been demonstrated from the significantly different dietary profile of patients at baseline, although patients from experimental group has surpassed control group in terms of their dietary quality after intervention.

Third, although sample size was not justified because it was a preliminary study, what would be the desirable sample size according to the study findings? Authors should also put this calculation in main text.

Fourth, to understand whether patients followed the dietary recommendation for HD patients, in additional to continuous variables, nutrient intake should be treated as categorical variables, as categorized by whether fulfilling dietary recommendations.

Nil

Author Response

We greatly appreciate your thoughtful comments that helped improve the manuscript.

We reflected your suggestion in the manuscript.

Please see the attachment, and find highlight the revised text using red colored highlighting. 

I look forward to your reply regarding our manuscript.

Reviewer 2 Report

Thank you for giving me the opportunity to review this paper.

Here some comments on the submitted manuscript:

- Study design (page #2, line #56): the authors wrote "This study was designed as a nonequivalent control group pretest-posttest design". Why did the authors choose this study design? It would be appropriate to provide a brief explanation.

- Did the authors registerd the study protocol?

- In the "General characteristics of the subjects", the authors wrote: "The participants were interviewed using a structured questionnaire". Which questionnaire have been used? Is it validated?

- page #5, line #127: "Twenty-four recall". I think the authors mean 'twenty-four-hour recall' or, better yet, 'twenty-four-hour dietary recall'.

- page #5, line #132:  (CAN Pro) 5.0. The company name and the State should be indeicated.

Author Response

(The authors gave the same response as above.)

Round 2

Reviewer 1 Report

My comments have been adequately addressed, I have no further comments.

Nil

Reviewer 2 Report

I thank the authors for having revised the manuscript in consideration of the provided comments.

The current form is suitable for publication.